# EpiSwitch PSE Blood Test Reduces Unnecessary Prostate Biopsies: A Real-World Clinical Utility Study

**DOI:** 10.3390/cancers17132193

**Published:** 2025-06-29

**Authors:** Joos Berghausen, Joe Abdo, Ryan Mathis, Ewan Hunter, Alexandre Akoulitchev, Garrett D. Pohlman

**Affiliations:** 1Department of Pharmacology and Physiology, Georgetown University Medical Center, 3900 Reservoir Rd NW, Washington, DC 20057, USA; jhb109@georgetown.edu; 2Oxford BioDynamics Inc., 7495 New Horizon Way, STE 110, Frederick, MD 21703, USA; joe.abdo@oxfordbiodynamics.com (J.A.); ryan.mathis@oxfordbiodynamics.com (R.M.); 3Oxford BioDynamics PLC, 3140 Rowan Palace, John Smith Drive, Oxford OX4 2WB, UK; ewan.hunter@oxfordbiodynamics.com (E.H.); alexandre.akoulitchev@oxfordbiodynamics.com (A.A.); 4Kearney Urology Center PC, 123 West 31st St., Kearney, NE 68847, USA

**Keywords:** prostate cancer detection, EpiSwitch PSE assay, 3D genome conformation, blood-based diagnostics, biopsy avoidance, real-world evidence, precision oncology

## Abstract

The EpiSwitch^®^ PSE blood test improves prostate cancer detection by reducing unnecessary biopsies by up to 79.1% in the clinical setting. It enhances clinical decision making, minimizes patient risk, and offers significant healthcare cost savings. Real-world evidence supports its adoption as a minimally invasive, cost-effective reflex test for improving early detection of prostate cancer and its use for high-risk groups in prostate cancer screening.

## 1. Introduction

Prostate cancer (PCa) remains a major global health burden and is one of the most commonly diagnosed malignancies in men. It is currently the second most frequently diagnosed cancer and the fifth leading cause of cancer-related death among men worldwide [1]. In 2018, the International Agency for Research on Cancer (IARC) reported approximately 1.3 million new cases globally, accounting for over 7% of all cancers in men [2]. This number is expected to rise sharply, with incidence rates projected to double by 2040 [3]. Furthermore, metastatic PCa is similarly on the rise [4]. Early-stage PCa can be readily treated, whereas in later stages that involve metastasis, the mortality rate is significantly increased [5,6]. Given its substantial burden on global healthcare and risk of mortality, early detection of PCa is of utmost importance.

Currently, the gold standard for detecting PCa is measuring prostate-specific antigen (PSA) levels [7]. However, both malignant and benign prostate disease can lead to an increase in PSA [8,9]. Even non-medical conditions, such as exercise [10] and ejaculation [11], have been shown to lead to an increase in PSA level. The clinical utility of traditional PSA screening has long been debated due to its limited specificity, which leads to a high rate of false-positive results and unnecessary biopsies. While PSA can be sensitive at lower thresholds, it lacks the precision needed to confidently distinguish malignant from benign causes of elevated levels. Ultimately, it has been estimated that 75% of patients with elevated PSA levels have no evidence of cancer, leading to unnecessary and invasive follow-up procedures [12].

A common follow-up procedure is multi-parametric magnetic resonance imaging (MP-MRI). This technique captures a detailed image of a patient’s prostate, which can be evaluated for PCa [13]. The PROMIS study shows that using MP-MRI after elevated PSA can reduce the number of unnecessary biopsies by 25% [14]. Despite representing a substantial advancement over PSA testing, there remains considerable room for further improvement to avoid false positives. Furthermore, MP-MRIs are associated with considerable cost, making the cost-to-benefit ratio highly debated [15,16].

The gold standard to confirm and determine the grade of prostate cancer are transrectal or transperineal biopsies [17]. These procedures can have significant side effects, including rectal bleeding, hematospermia, erectile dysfunction, pain, infection, and urinary retention [18,19]. While prostate cancer biopsies play a crucial role in diagnosis and staging, the high false-positive rate associated with PSA testing has led to an estimated 750,000 unnecessary biopsies per year [20]. This leads to a significant drain on the healthcare system and a financial burden on patients. These issues highlight the need for more accurate diagnostic approaches to reduce the number of unnecessary biopsies and to ensure that only patients with a true risk of prostate cancer undergo these invasive assessments and are diagnosed.

EpiSwitch PSE is a novel test that analyzes epigenetic markers, called circulating chromosome confirmation signatures, in whole blood samples to determine if a patient has prostate cancer. These signatures are maintained through cell division and retain essential regulatory and environmental information that shapes an individual’s phenotype [21,22,23]. The test has previously demonstrated potential to guide clinical decisions with a high level of agreement with biopsy outcomes, offering meaningful risk stratification in real-world settings, ultimately leading to a significant reduction in unnecessary biopsies [22,24].

Three-dimensional (3D) genome conformation mapping is a molecular technique that captures the spatial organization of chromatin within the nucleus, revealing physical interactions between genomic regions that are often far apart in linear sequence. These long-range interactions play key roles in gene regulation and are sensitive to changes in cellular state, including malignant transformation. Aberrant 3D genomic architecture has been implicated in cancer development and progression, influencing oncogene activation, tumor suppressor silencing, and epigenetic dysregulation [25]. Prior studies have demonstrated the clinical utility of 3D genomic signatures in detecting and stratifying cancer risk across multiple tumor types, including prostate cancer, through the identification of disease-specific chromatin interaction patterns [21,26,27].

In this study, we conducted a real-world observational analysis of patients who received the EpiSwitch PSE test to evaluate its clinical utility, economic impact, and role in shared decision making. Specifically, we assessed how PSE results influenced biopsy decisions, reduced the number of unnecessary invasive procedures, improved communication between physicians and patients, and supported more personalized, evidence-based care planning. We also estimated potential cost savings associated with reduced imaging and biopsy utilization, highlighting the broader healthcare system benefits of integrating PSE into prostate cancer diagnostic workflows.

PSA testing remains a cornerstone of early prostate cancer detection, yet it suffers from limited specificity, leading to a high number of negative biopsies and considerable patient burden. Both transrectal and transperineal prostate biopsies carry risks, including infection, bleeding, urinary complications, and patient anxiety, making the need for a more accurate, non-invasive triage tool increasingly urgent.

In this study, we hypothesized that the EpiSwitch PSE assay, which detects cancer-specific 3D genome conformational signatures from blood, could improve diagnostic precision by identifying patients at low or high risk of harboring clinically significant prostate cancer. Our primary goal was to evaluate whether PSE could help reduce unnecessary biopsies without compromising safety, thereby decreasing associated morbidity and lowering healthcare costs through more efficient resource utilization.

## 2. Materials and Methods

### 2.1. Study Design

This was an observational real-world evidence study designed to evaluate the clinical utility of the EpiSwitch^®^ PSE blood-based assay in identifying patients’ likelihood of having prostate cancer. The study reviewed patient records from a single institution, capturing those who underwent both standard PSA screening and EpiSwitch PSE testing between October 2023 and May 2025. All patients included had a minimum of one month of clinical follow-up, with outcome data derived from subsequent imaging (e.g., MRI) and/or biopsy, where applicable. No interventions were administered for study purposes, and data were collected solely through chart review in compliance with HIPAA regulations.

### 2.2. IRB or Ethics Approval

This observational real-world study and medical chart analysis were conducted in accordance with the principles outlined in the Declaration of Helsinki. Institutional Review Board (IRB) approval was obtained from WIRB-Copernicus Group IRB (Puyallup, WA, USA); Study WO#: 1-1852637-1. Given the retrospective nature of the study and the use of deidentified patient data, the requirement for informed consent was waived.

### 2.3. Patient Population

A total of 187 male patients aged 45–79 years were included. Eligibility criteria required PSA > 3 ng/mL and/or an abnormal digital rectal exam (DRE), with exclusion of patients over 80 and those with a current or prior prostate cancer diagnosis, history of prostatectomy, or any serious condition precluding clinical evaluation. Among the cohort, 53 patients had biopsy-confirmed diagnoses, forming the training set, while the remaining 134 were assessed using predictive modeling (Table 1 and Table 2). Patients with a prior diagnosis of prostate cancer or incomplete clinical data were excluded to ensure data integrity.

### 2.4. Sample Size Consideration:

As this was a retrospective observational study based on real-world clinical data, no formal sample size calculation was performed in advance. The study included all eligible patients who underwent EpiSwitch PSE testing between October 2023 and May 2025 at the study site, resulting in a total sample size of 187 patients. This approach ensured that the analysis reflected the full spectrum of clinical practice during the study period.

### 2.5. EpiSwitch PSE Overview

EpiSwitch PSE utilizes proprietary 3D genome conformational profiling technology to detect spatial chromatin interaction patterns associated with prostate cancer risk. Following DNA extraction and chromatin preparation, specific long-range interactions are captured, amplified, and quantified using PCR-based detection. The resulting signal profiles are interpreted using a machine learning algorithm trained to classify cancer risk. Full analytical validation of the assay workflow, including sensitivity, specificity, and reproducibility, has been described previously [22].

### 2.6. EpiSwitch PSE Testing Procedure

The EpiSwitch PSE assay is a blood-based test that evaluates 3D genomic conformational biomarkers to stratify patients by likelihood of having prostate cancer. High accuracy of 94% for non-invasive early detection of prostate cancer by EpiSwitch PSE have been demonstrated in the PROSTAGRAM trial by Imperial NHS Trust [22]. The results are reported as binary outputs, namely low likelihood of prostate cancer (LLPC) or high likelihood of prostate cancer (HLPC).

Peripheral blood samples were collected using standard venipuncture into K2-EDTA tubes and gently inverted to mix. Samples were stored at room temperature and shipped overnight to the laboratory with no need for controlled temperature conditions. All samples were processed within 48 h of collection to preserve chromatin integrity. Upon arrival, samples underwent quality control assessment prior to 3D genome conformational profiling using the EpiSwitch PSE assay. The entire process, from blood draw to final assay readout, followed standardized operating procedures in a CLIA-certified laboratory (OBD Inc., Frederick, MD, USA) under standard operating conditions and USA reimbursement code CPT PLA 0433U. In the United Kingdon, BUPA UK covers PSE tests run by UKAS-accredited clinical laboratory (Oxford, UK, OBD plc). Test reports with the PSE results were generated automatically, with calls independent of biopsy or imaging data.

### 2.7. Data Collection

Demographic and clinical data were extracted from electronic medical records and included age, PSA levels, DRE findings, PSE results, MRI and biopsy reports, and clinical follow-up notes. Imaging modalities (e.g., PI-RADS MRI scores) and biopsy histopathology (e.g., Gleason grades) were also documented. Decision pathways regarding whether to proceed with biopsy following PSE testing were noted where available.

### 2.8. Outcome Measures

The primary outcome measures of this study centered on evaluating the clinical utility of the EpiSwitch PSE test, with particular emphasis on its ability to reduce unnecessary prostate biopsies, streamline diagnostic workflows, and support shared decision-making between patients and clinicians. A key goal was to assess how the binary high/low likelihood results from the PSE test influenced patient management, particularly in deferring invasive procedures when a low likelihood result was obtained. These outcomes were closely linked to improvements in communication around the next steps, allowing clinicians to offer patients more personalized and less burdensome diagnostic pathways. Secondary analyses evaluated the concordance between EpiSwitch PSE results and clinical outcomes, including agreement between test classifications and confirmed diagnoses or absence of disease via MRI or biopsy. The study also estimated potential healthcare cost savings associated with reduced reliance on invasive diagnostics, further underscoring the test’s role in improving efficiency and lowering the burden on patients and healthcare systems.

### 2.9. Statistical Analysis

Predictive modeling was performed using XGBoost (version 1.7.7.1), a gradient-boosted decision tree algorithm. A training set consisting of 53 biopsy-confirmed patients was used to develop the models, with hyperparameter tuning conducted to optimize performance. Model evaluation utilized 10-fold cross-validation to assess generalizability and avoid overfitting. SHAP (SHapley Additive exPlanations) analysis was applied to interpret feature importance within the models. Key clinical variables, including EpiSwitch PSE results and PI-RADS scores, were incorporated into the models (Table 3). Model performance was evaluated based on alignment with observed clinical decisions and downstream patient management outcomes, which were reported with 95% confidence intervals where appropriate.

Two predictive models were developed using the cohort of 53 biopsy-confirmed patients and then applied to predict biopsy outcomes in the remaining 134 patients who had not undergone biopsy. The primary difference between the models lies in how they classified Gleason 3 + 3 prostate disease. Model B treated Gleason 6 as clinically significant cancer, while Model A considered Gleason 3 + 3 as clinically indolent (low-grade) disease aligning with some contemporary perspectives that categorize this disease as low-risk and potentially suitable for active surveillance. This difference in classification criteria directly influenced each model’s threshold for determining who could safely avoid biopsy.

## 3. Results

### 3.1. Patient Demographics

A total of 187 male patients were included in the study (Table 1). All patients underwent evaluation for suspected prostate cancer between October 2023 and March 2025 based on elevated PSA levels (>3 ng/mL) and/or abnormal digital rectal exam (DRE) findings. The median age across the cohort was 64 years (range: 45–79 years). Among these, 53 patients underwent confirmatory prostate biopsies, forming the training cohort for predictive model development, while 134 patients were evaluated using predictive modeling based on EpiSwitch PSE results and clinical parameters (Table 2).

PSA levels among high likelihood (HLPC) patients (n = 88) averaged 9.1 ng/mL (median 8.2 ng/mL, range 4.4–23.6 ng/mL), while low likelihood (LLPC) patients (n = 99) showed a mean PSA of 6.1 ng/mL (median 5.3 ng/mL, range 1.3–29.9 ng/mL). There was no meaningful difference in age distribution between the HLPC and LLPC groups.

### 3.2. Clinical Decision Impact

In this study, low-likelihood results from the PSE test played a role in guiding clinical decisions about whether patients should undergo a biopsy. In cases where the PSE result suggested a low probability of PCa, clinicians and patients almost always opted to defer or avoid biopsy. As a result, biopsy data were unavailable for a subset of patients, introducing missing outcomes into the dataset. To address this limitation, cross-validation techniques were employed to predict the testing accuracy of PSE on the missing data using the complete dataset.

Using the EpiSwitch PSE blood test as a triage tool, predictive models indicated that approximately 79.1% of patients could potentially avoid unnecessary prostate biopsies.

Predictive models incorporated EpiSwitch PSE test results alongside relevant clinical parameters. Among these, Model A demonstrated the highest performance, predicting that 79.1% (106 out of 134) of patients could safely avoid unnecessary prostate biopsy based on a negative classification. Model A treated Gleason 3 + 3 as clinically indolent prostate disease. We also developed Model B that treated 3 + 3 prostate histology as clinically significant prostate disease. Model B showed that 66.4% (89 out of 134) of patients could avoid biopsy (Table 2).

These findings consistently highlight the strong potential of the EpiSwitch PSE assay to act as an effective triage tool, substantially reducing the number of invasive diagnostic procedures performed. Internal model validation using 10-fold cross-validation demonstrated stable performance metrics across iterations, underscoring the robustness and clinical applicability of the predictive framework developed in this study.

### 3.3. Workflow Efficiency and Economic Impact

Operationally, the assay demonstrated a 100% technical success rate, with all 187 samples processed yielding valid results. The average turnaround time (TAT) for PSE testing was 4.4 days, supporting its feasibility for real-world clinical use.

Based on predictive modeling and biopsy outcomes, the PSE assay potentially avoided approximately 97 unnecessary prostate biopsies and 95 MRIs in this cohort of patients who never required a biopsy based on the study protocol. This translates into an estimated cost avoidance of over USD 230,000, accounting for reduced procedural volume, imaging utilization, and downstream complications, such as hospitalizations, due to biopsy-related adverse events. The healthcare system could realize an average savings of approximately USD 1275 per patient. These data underscore the economic value of incorporating PSE into the prostate cancer diagnostic pathway by improving triage precision, optimizing resource use, and reducing procedural risk.

When extrapolated to the national level, the healthcare economic impact of incorporating PSE into prostate cancer screening workflows could be substantial. An estimated 1 million prostate biopsies are performed annually in the U.S., with as many as 75% deemed potentially unnecessary [20]. By more precisely stratifying patients prior to biopsy, PSE has the potential to help avoid up to 593,000 procedures per year, conservatively. At an estimated average cost of USD 2500 per biopsy, including procedural, pathology, and complication-related expenses, this represents a potential annual savings of approximately USD 1.48 billion. When combined with additional cost savings from avoided MRIs and follow-up care, the total economic benefit could approach USD 2 billion annually. These figures highlight the broader systemic value of deploying blood-based risk stratification tools like PSE in routine prostate cancer diagnostics.

### 3.4. Feature Importance and Model Interpretation

To better understand the drivers of model predictions, SHAP (Shapley additive explanations) analysis was performed across both predictive models. SHAP values quantify the contribution of each input feature to the final model output, offering transparent insight into how specific clinical and molecular variables influenced individual patient classifications [28]. The initial data split was based on the PSE outcome, and analysis across Models A and B consistently identified the EpiSwitch PSE blood test result as the most influential predictor of biopsy outcome (Figure 1 and Figure 2). This highlights that initial guidance via the PSE outcome not only drove the success of the models but also enabled clearer interpretation of other clinical variables, reinforcing the central role of PSE in supporting clinical decision making.

In parallel, the PI-RADS score from MRI imaging was identified as another major contributing factor, with higher scores favoring a positive biopsy prediction. Other features, such as PSA levels, patient age, and previous 4Kscore^®^ test (OPKO Labs, Elmwood Park, NJ, USA) results, showed a secondary but meaningful influence on model behavior (Figure 1 and Figure 2). Importantly, in all models, a low-likelihood PSE result heavily shifted predictions toward a negative classification, reinforcing the assay’s strength as a non-invasive indicator of reduced cancer risk (Table 3).

The consistency of feature importance rankings across different model architectures strengthens confidence in the biological relevance and clinical utility of the PSE assay. Furthermore, SHAP analysis revealed that the models maintained logical, clinically coherent relationships between input features and outcomes, avoiding erratic or biologically implausible patterns often seen with overfitted or poorly generalized models. This interpretability supports potential real-world adoption, as clinicians can better understand and trust model recommendations when they align with known prostate cancer risk factors. The combination of high model transparency, biological plausibility, and clinical relevance positions EpiSwitch PSE as a robust tool to enhance prostate cancer diagnostic workflows.

### 3.5. Cases Demonstrating Additional PSE Utility

Three illustrative case studies underscore the clinical utility of EpiSwitch PSE in informing prostate cancer management. In Case 1, a patient received a low-likelihood PSE result despite a suspicious PI-RADS 5 lesion observed via MRI. A subsequent MRI/TRUS fusion-guided biopsy was negative for prostate cancer, supporting the reliability of the PSE result and helping to avoid overtreatment. In Case 2, the PSE test indicated a high likelihood of cancer, but, due to the presence of a cardiac pacemaker, MRI was contraindicated. The urologist proceeded directly to biopsy, which confirmed clinically significant prostate cancer. In Case 3, a patient with a high-likelihood PSE result had a negative MRI; however, guided by the PSE result, the treating urologist and patient opted for a systemic 12-core biopsy. All 12 cores revealed Gleason 5 + 4 = 9 lesions, indicating aggressive disease that would have otherwise been missed by imaging alone. These cases demonstrate how PSE can complement or even override imaging findings, enhancing diagnostic confidence and enabling timely, personalized clinical decision-making.

## 4. Discussion

Our real-world evidence study demonstrates that the EpiSwitch PSE blood-based assay can meaningfully improve prostate cancer detection pathways by stratifying patients according to cancer risk. Patients with a low PSE result never harbored clinically significant prostate cancer in our cohort. This observation underscores the assay’s high negative predictive value and potential as a reliable “rule-out” test. Notably, prior validation studies of the PSE assay reported aligns with our real-world findings that low PSE calls are strongly associated with absence of prostate cancer [29]. By accurately identifying patients unlikely to have prostate cancer, the PSE assay addresses a critical gap in the current diagnostic paradigm and lays the foundation for significant clinical utility.

A recently published review article summarized the clinical performance of several non-invasive biomarker tests, including PHI, 4Kscore, and IsoPSA [24], providing valuable context for evaluating the PSE test. While direct head-to-head comparisons remain limited, the review indicates that the PSE test demonstrates improved biostatistical accuracy. Furthermore, the PSE test offers advantages in terms of ease of implementation within clinical workflows and applicability across diverse patient populations, supporting its potential utility as a complementary diagnostic tool.

### 4.1. Clinical Utility and Patient Impact

The integration of the EpiSwitch PSE assay into prostate cancer diagnostic pathways could substantially enhance clinical decision making and patient care. Today’s standard practice often relies on prostate-specific antigen (PSA) screening, which is notoriously low in specificity, leading to a high rate of unnecessary biopsies [30,31]. Many men with elevated PSA undergo an invasive biopsy only to discover they do not have clinically significant cancer. This exposes patients to procedure-related discomfort, anxiety, and potential complications—all for little clinical benefit. Our findings suggest that using PSE as a reflex test or pre-biopsy triage can dramatically reduce this burden. In our real-world cohort, roughly 80% of patients (106 out of 134) could have safely avoided an immediate biopsy based on a low PSE result, sparing them the pain and risks of an unnecessary procedure without compromising cancer detection (Figure 3). Reducing the biopsy volume in this way directly translates into improved patient experiences and more streamlined care pathways. Fewer men will experience biopsy-related side effects (such as bleeding, infection, or hospitalizations), and fewer indolent, low-grade tumors will be incidentally detected and overtreated.

Overall, the PSE assay’s ability to rule out significant cancer in low-risk patients means that clinicians can confidently defer or forego biopsy in those cases, focusing attention on patients who truly need further intervention (Table 3). This level of clinical utility elevates prostate cancer care by minimizing harm while maintaining diagnostic accuracy. Notably, similar biomarker-driven approaches in prostate cancer diagnostics have reported substantial reductions in unnecessary biopsies, on the order of 20–50% [30]. Our results with the PSE test, namely an approximately 79.1% reduction in biopsy procedures, demonstrate a significant improvement, reinforcing that PSE provides a competitive advantage in patient stratification better than other emerging tests. In contrast to complex or resource-intensive tools (for example, MRI or elaborate molecular panels), PSE offers these benefits with a simple blood draw and rapid turnaround, making it especially attractive for broad clinical adoption.

### 4.2. Health Economic Implications

Beyond the clear clinical benefits, our study highlights significant health economic outcomes research (HEOR) advantages associated with the PSE assay. Avoiding unnecessary biopsies has a direct economic impact on healthcare systems. Each prostate biopsy carries direct costs (procedure, anesthesia, pathology analysis) and indirect costs (management of side effects, time off work, patient distress). When PSA screening prompts a high volume of biopsies with low yield, the cumulative cost burden is substantial. Indeed, the low specificity of PSA is estimated to result in millions of dollars in avoidable medical expenditures due to unnecessary procedures and indirect costs [30]. By introducing PSE as a triage test, the number of unwarranted biopsies can be dramatically curtailed. Our real-world data indicates a nearly 80% reduction in biopsy procedures when PSE results are factored into the decision process. In practical terms, this could translate to enormous cost savings. Fewer biopsies mean lower utilization of operating room time or clinic resources, fewer pathology evaluations, and fewer treatments of biopsy-related complications. From the payer perspective, a blood test that drastically reduces the biopsy rate for equivocal PSA findings would improve the cost-effectiveness of prostate cancer screening and diagnosis. For example, a recent cost-effectiveness analysis of the PHI biomarker test demonstrated that reducing unnecessary biopsies by roughly 20–50% yielded significant healthcare savings and added quality-adjusted life years, making the approach economically favorable [30]. In our scenario, PSE’s 66.4–79.1% biopsy avoidance rate indicates an even greater economic benefit. Moreover, beyond direct cost savings, there are system-level efficiencies gained: urology clinics could avoid backlogs by not performing low-yield biopsies, and physicians could allocate time and resources to patients with positive PSE (high-risk) results who need further evaluation. This aligns with value-based care principles, where interventions that reduce waste and improve patient outcomes are highly sought after. The PSE assay, therefore, has the potential not only to improve individual patient management but also to favorably shift health economics by lowering unnecessary healthcare utilization. In summary, the use of EpiSwitch PSE can help bend the cost curve in prostate cancer care, delivering better health outcomes at lower cost—a mutual benefit for patients, providers, and payers alike [32].

### 4.3. Comprehensive and Shared Decision-Making

An additional strength of the EpiSwitch PSE assay observed in routine clinical practice is its binary result format, classifying patients as having either a high or low likelihood of prostate cancer, which markedly improves communication with patients. Compared to other risk assessment tools that present numeric or percentage probabilities as risk scores, the PSE result is more intuitive and easier for patients to understand. This clarity simplifies shared decision making and facilitates more confident choices regarding the next step, whether that be observation, further imaging, or biopsy. Every patient presented with the PSE option opted to proceed with testing, underscoring a strong patient preference for additional molecular information to guide their care. Notably, when offered multiple management pathways, including immediate biopsy, MRI, or a combination, most patients preferred to first obtain the PSE result alone and defer further intervention unless indicated. Only a minority chose to proceed with MRI concurrently. These findings suggest that the PSE test empowers patients by providing accessible, actionable results, and may enhance satisfaction and engagement in prostate cancer care.

### 4.4. Clinical Utility Beyond Avoiding Unnecessary Biopsies

The case studies underscore the broader clinical value of the EpiSwitch PSE test in supporting prostate cancer diagnostic pathways. By providing a binary risk classification, the test enhances physician confidence in guiding next steps, whether those involves deferring invasive procedures, proceeding directly to biopsy, or acting despite inconclusive imaging. Importantly, PSE offers a non-invasive alternative in cases where MRI is contraindicated, such as patients with implanted cardiac devices, making these individuals strong candidates for PSE-based triage. The test can also be administered to any male regardless of PSA level, digital rectal exam (DRE) timing or findings, or the presence of benign prostatic hyperplasia (BPH), including those currently on medications for BPH. This versatility makes PSE a viable and practical option across a wide range of clinical scenarios involving prostate health.

Notably, in one case, PSE flagged high risk in a patient with a negative MRI, ultimately leading to the detection of a Gleason 9 prostate cancer, an outcome that may well have been lifesaving. Beyond its adaptability, PSE supports shared decision making, streamlines diagnostic workflows, and enables more personalized patient management (Figure 4). Overall, its integration into routine clinical practice has the potential to improve outcomes, reduce unnecessary interventions, and fill critical gaps when traditional diagnostic tools are limited or unavailable.

### 4.5. Technical Success

The 100% technical success rate achieved in this study is a critical strength, underscoring the robustness and reliability of the EpiSwitch PSE assay in real-world clinical settings. Consistent technical performance ensures that results are reproducible and reduces the risk of sample processing failures that could delay diagnosis or require repeat testing. In addition, the assay’s rapid average turnaround time of 4.5 days further enhances its clinical utility, enabling timely decision making for patients and providers. Together, high technical fidelity and quick turnaround time support broader clinical adoption by minimizing operational barriers. This provides greater confidence in the PSE’s dependability across a variety of healthcare settings.

### 4.6. Study Limitations

This analysis was conducted on data from a single tertiary center in the Midwest, USA, which may introduce selection bias and limit generalizability. Biopsy decisions in our cohort were influenced by PSE results in real time, meaning that most men with Low PSE were not biopsied, and, thus, any missed cancers in that group are inferred rather than confirmed. These factors underscore the need for cautious interpretation of negative findings. However, the very nature of this real-world evidence design is also a strength as it reflects everyday clinical practice and physician behavior when armed with a new test.

To further address the potential for confirmation and selection biases inherent in this real-world design, we applied cross-validation techniques to estimate test performance among patients who were not biopsied. By leveraging patterns observed across the full dataset, this approach allowed us to infer the likely diagnostic accuracy of PSE in cases where histopathologic confirmation was unavailable. While not a substitute for biopsy, this statistical mitigation strengthens confidence in the test’s negative predictive performance and supports the robustness of our findings despite incomplete verification.

Our models were trained and internally validated using data from a single-center cohort, which may limit its generalizability to broader populations. Although we employed 10-fold cross-validation and regularization techniques to minimize overfitting, the model’s performance should be interpreted with caution. Future work should include external validation using multicenter datasets and prospective studies to assess the model’s performance across diverse clinical settings and patient populations.

### 4.7. Future Directions

Building on the promising results of this study, future research could focus on prospective and longitudinal evaluations of the PSE assay to further establish its role in the various settings of prostate cancer care. A multicenter prospective study is a logical next step to confirm our findings across diverse populations and healthcare settings. This would allow for a head-to-head assessment of outcomes, for instance, comparing biopsy rates and cancer detection rates between patients managed with PSE guidance and those managed by standard criteria. Endpoints could include a reduction in biopsy procedures, the detection of clinically significant cancers, and any cases of missed high-grade disease. In parallel, longitudinal surveillance research is warranted to explore the utility of PSE in monitoring patients over extended time durations. One application could be in men who defer biopsy due to a low PSE result—periodic retesting with PSE (every 6–12 months, for example) could be studied as a strategy for early warning if a patient’s risk status changes. This longitudinal approach would clarify how PSE performs in an active surveillance or screening context, ensuring that cancers are not missed long-term while still minimizing interventions.

The EpiSwitch PSE blood-based assay has shown significant promise in the real-world clinical setting, and with further usage, it could herald a new standard of precision in prostate cancer diagnostics, one that optimizes patient outcomes while reducing unnecessary procedures and costs.

## 5. Conclusions

This real-world evidence study demonstrates that the EpiSwitch PSE blood-based assay offers a significant advancement in the early detection and risk stratification of prostate cancer. By integrating 3D genome conformational profiling into clinical evaluation, the assay enables more precise, non-invasive identification of patients at risk, thereby supporting informed clinical decision-making. Depending on the model, up to four out of five patients (66.4% to 79.1%) may be eligible to defer biopsy without harm, representing a substantial improvement in diagnostic efficiency.

The ability to confidently defer biopsy in low-risk patients not only minimizes exposure to invasive procedures and potential complications but also reduces emotional distress and healthcare costs. The integration of PSE results with standard imaging and laboratory data enhances the utility of current diagnostic pathways, yielding a more personalized and economically sustainable approach to prostate cancer screening.

While these findings are encouraging, this study has several limitations. It was conducted at a single site, and the observational design may introduce selection bias, particularly among patients who did not proceed to biopsy. Additionally, because data were collected retrospectively and deidentified, some clinical variables could not be fully controlled. The study was approved by an independent IRB, and all ethical standards for retrospective research involving human subjects were followed.

In summation, these findings support broader adoption of the EpiSwitch PSE assay in urology clinics and warrant continued clinical adoption and further real-world evaluation across diverse healthcare settings.

## Figures and Tables

**Figure 1 cancers-17-02193-f001:**
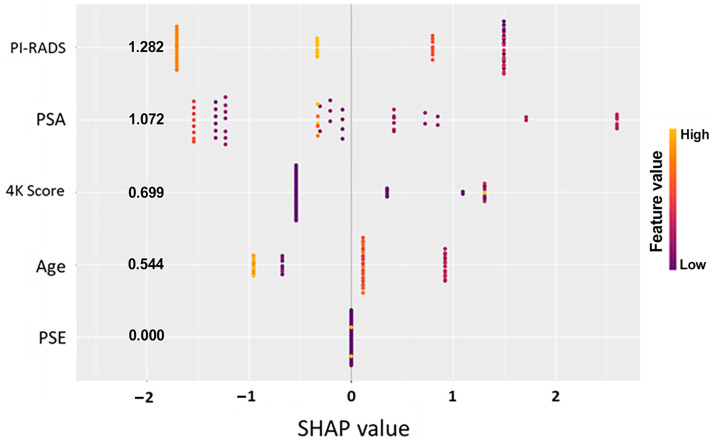
SHAP summary plot illustrating feature importance in the predictive model for prostate cancer risk. Each dot represents a single patient, with the position on the x-axis indicating the SHAP value (impact on model output) for that feature. Features are ranked by mean absolute SHAP value, with higher values indicating greater influence on the model’s prediction. Color represents the original value of the feature (purple = low, yellow = high). In this model, the binary EpiSwitch PSE result had the strongest impact on predicted risk when combined with MRI pi_rads, showing that 79.1% of patients would be saved from biopsies.

**Figure 2 cancers-17-02193-f002:**
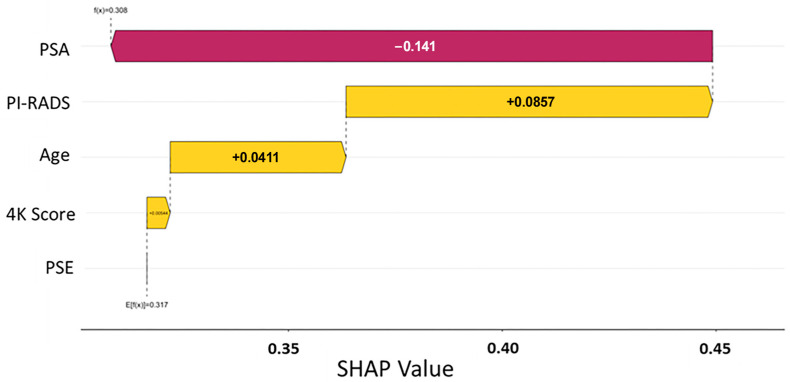
SHAP summary plot illustrating feature importance in the predictive model for prostate cancer risk. This plot illustrates how each input variable influenced the model’s output prediction relative to the baseline (expected value). The base value of 1.2 represents the model’s expected prediction across the dataset, while the final output of 0.988 reflects the individualized prediction for the patient. Negative SHAP values (shown in red) decrease the predicted risk, whereas positive values (shown in yellow) increase it. In this example, a high PSA level contributed most to lowering the predicted risk (−0.141), while the PI-RADS slightly increased it (+0.086). Other features, such as 4K score, age, and PSE, had smaller marginal effects. This individualized explanation supports interpretability of the model’s output and reinforces trust in risk stratification.

**Figure 3 cancers-17-02193-f003:**
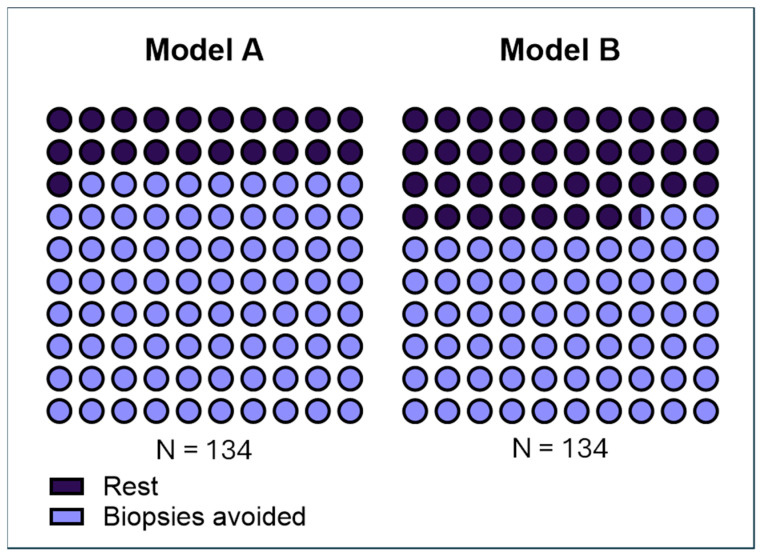
Scaled icon array of biopsy avoidance rate of models A and B. Displayed is the predicted biopsy avoidance rate of models A and B if PSE was used as a triage tool. Each graph displays 100 icons, with each icon representing approximately 1% of the 134-patient cohort. Blue icons indicate the proportion of patients predicted to avoid biopsy, and black icons represent the rest of patients who would continue down PCa screening pathways. This figure highlights the differential impact of the two models on clinical decision-making pathways depending on how the clinician classifies Gleason 3 + 3.

**Figure 4 cancers-17-02193-f004:**
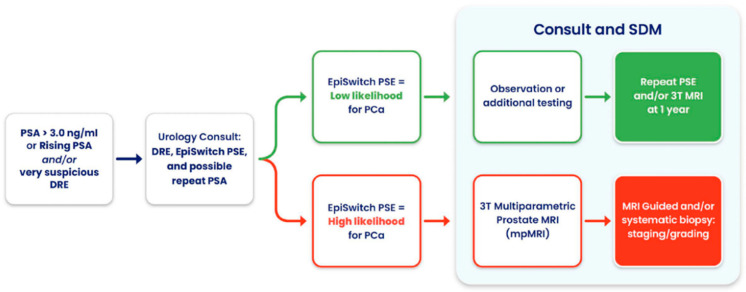
Patient workflow under prostate cancer surveillance incorporating PSE: Illustration of a clinical decision-making pathway integrating the EpiSwitch^®^ PSE blood test for patients with elevated PSA or abnormal DRE findings. Based on the PSE result, patients are stratified into low or high likelihood of prostate cancer (PCa). Those with a low-likelihood result may be observed or undergo additional non-invasive testing, with reevaluation at one year. Patients with a high-likelihood result are referred for multiparametric MRI (mpMRI) and, if indicated, MRI-guided or systematic prostate biopsy. This pathway supports shared decision making and more efficient, personalized care.

**Table 1 cancers-17-02193-t001:** Patient demographics. Baseline demographic and clinical characteristics of the 187 patients included in the retrospective real-world analysis of the EpiSwitch PSE assay. The cohort consisted of 53 biopsy-confirmed patients used for model training and 134 patients without biopsies used for clinical utility testing. All data were collected between 2024 and 2025 from routine urology practice settings.

Characteristic	Value
Total patients	187
Median age (years)	64
Age range (years)	45–79
PSA level range (ng/mL)	3–30
Biopsy-confirmed patients (training cohort)	53
Patients without biopsies (testing Cohort)	134

**Table 2 cancers-17-02193-t002:** Performance of predictive models in identifying patients who could safely avoid transrectal or transperineal prostate biopsies. Performance comparison of two predictive models estimating which patients could safely avoid prostate biopsy based on EpiSwitch PSE results and clinical variables. Both models were applied to the same cohort of 134 patients. Model A classified only Gleason ≥ 3 + 4 as malignant, while Model B included Gleason 3 + 3. Biopsy avoidance rate and model accuracy are shown for each approach.

Model	Gleason Score Classified as Malignant	Total Patients Tested (n)	Model Accuracy	Patients Predicted to Avoid Biopsy (n)	Biopsy Avoidance Rate (%)
Model A	3 + 4	134	77.3%	106	79.1
Model B	3 + 3	134	80.0%	84	62.7

**Table 3 cancers-17-02193-t003:** SHAP feature importance ranking. Relative importance of clinical and molecular features in the predictive model, ranked by SHAP (Shapley additive explanations) values. Higher-ranked features contributed more substantially to individual model predictions, with the EpiSwitch PSE result showing the greatest overall influence. Rankings are based on SHAP analysis of the final XGBoost model trained on a real-world cohort of 134 patients undergoing prostate cancer evaluation.

Feature	Relative Importance
EpiSwitch^®^ PSE result	Highest
PI-RADS score	High
PSA level	Moderate
Patient age	Moderate
4Kscore^®^ test result	Lower

## Data Availability

The data supporting the findings of this study are available from the corresponding author upon reasonable request. Due to patient privacy and institutional policies, individual-level clinical data are not publicly accessible.

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
