# Peer review of "EpiSwitch PSE Blood Test Reduces Unnecessary Prostate Biopsies: A Real-World Clinical Utility Study"

_cancers, 2025, doi:10.3390/cancers17132193_

Round 1

Reviewer 1 Report

Comments and Suggestions for Authors

The work titled " Real-world observational study of the clinical utility of blood-based 3D genome conformation mapping (EpiSwitch PSE) to reduce unnecessary biopsies and augment patient management strategies for early prostate cancer detection" represents a significant advancement in the field of cancer.

In this study, the authors were aimed to investigate the evaluation of the EpiSwitch® PSE assay, a blood-based test analyzing 3D genome conformation signatures, in 187 patients undergoing evaluation for PCa. Regarding the results about 79.1% of patients could safely defer biopsy based on PSE data, highlighting the test’s potential to reduce invasive procedures without compromising diagnostic safety. The PSE result, whether high or low, consistently emerged, alongside the PI-RADS score, as the strongest predictor of cancer risk, highlighting the PSE score as the primary driver of this outcome. Incorporating the EpiSwitch PSE assay into clinical workflows enhances decision-making efficiency, reduces unnecessary biopsies, and improves healthcare resource utilization. The findings of this work were support the assay’s strong clinical utility and economic value, highlighting its potential for broader adoption as a minimally invasive reflex test and a pre-biopsy triage tool for prostate cancer early and accurate detection. All of these findings seem valuable and suitable contribution to be published in the Cancers journal after justifying some points:

  • It is recommended to update the title to be simpler and more attractive for the readers and researchers
  • The results in the abstract section should be clearer with more findings’ values
  • You should remove PhD and MD from the authors name list
  • It is recommended to add a conclusion to the abstract with planed of the future work and recommendations
  • An abbreviations list should be added in the end of this work sections after conclusion
  • Control the references in the main text writing style
  • It is recommended to add more similar works to the introduction and discussion section to increase the number of references
  • The aim of this study lines 85-87 should be explained more
  • No need to write a separate section for IRB or ethical approval you can mention them in the other method section and in the end of this study in the declarations
  • The sample size calculation should be mentioned in the method section
  • Did the authors make the validation test before start collecting the data
  • The conclusion should be improved more by adding limitation and ethical considerations

Best wishes

Author Response

Please see the attached PDF for a detailed point-by-point response to each reviewer comment, including how we have addressed and incorporated the suggested revisions into the manuscript.

Reviewer 2 Report

Comments and Suggestions for Authors

This study presents compelling evidence for the utility of the EpiSwitch PSE assay as a minimally invasive blood test for prostate cancer detection. The findings demonstrate a significant reduction in unnecessary biopsies (up to 79.1%), which could have substantial implications for clinical practice by enhancing patient safety and reducing healthcare costs.

Comments: Minor Revision

  • The introduction would benefit from a more detailed explanation of the limitations associated with existing molecular and imaging biomarkers, such as their predictive accuracy or practical challenges, to better underline the advantages of the proposed assay.
  • While the concept of 3D genome conformation mapping is intriguing, a brief overview of its biological basis and prior evidence supporting its relevance to cancer detection would help readers unfamiliar with the technology.
  • The study's goals and the specific hypotheses or research questions are not explicitly stated. Clarifying these would strengthen the scientific framing and guide the reader through the study’s objectives.
  • Including a concise summary of prior validation studies or preliminary data supporting the efficacy of EpiSwitch PSE would bolster the justification for this investigation.
  • The description of sample selection criteria and patient demographics is limited in the provided extract; details such as inclusion/exclusion criteria, age ranges, PSA levels, ethnicity, comorbidities, and prior biopsy history should be explicitly stated for clarity and reproducibility.
  • The process of sample collection, handling, and processing for the EpiSwitch PSE assay lacks sufficient detail. For example, information on blood collection procedures, storage conditions, and processing timelines is essential for reproducibility.
  • The assay’s technical workflow, including specifics on the 3D genome conformational analysis, detection methods, and quality control procedures, is not described in detail. Providing a brief overview or referencing prior validation work would strengthen understanding of the assay’s reliability.
  • The performance metrics of the assay (e.g., sensitivity, specificity, positive/negative predictive values) are not explicitly mentioned, although some are implied via model performance; clear reporting of these metrics and their calculation methods would be beneficial.
  • A list of abbreviations should be added to the manuscript

Author Response

(The authors gave the same response as above.)

Reviewer 3 Report

Comments and Suggestions for Authors

The manuscript discusses the topic of improving early detection and patient management in prostate cancer using a new blood-based assay, the EpiSwitch PSE test, which evaluates 3D genome conformation. The methodology is described as generally sound, and the results are presented with clarity, including case studies and statistical modeling using SHAP and XGBoost.

However, there are still some issues that require clarification before publication:

  1. The study heavily relies on real-world, retrospective data with many decisions influenced in real-time by the test results. While this mimics clinical practice, it also introduces potential confirmation and selection biases. The authors should more explicitly discuss this in the limitations and outline how they mitigated bias, especially considering that most patients with Low PSE were not biopsied.
  2. The predictive model was trained and validated using internal cross-validation from a single-center cohort. A more detailed justification for the model’s robustness and how its findings might translate to broader populations should be included. A suggestion for external validation or prospective studies should be highlighted more prominently in the discussion.
  3. Although IRB approval and waiver of consent are mentioned, the ethical implications of using predictive models for deferring invasive procedures in a non-consented real-world setting should be more thoroughly discussed to reassure readers about patient autonomy and safety.
  4. The manuscript would benefit from a clearer schematic or decision tree summarizing how the PSE test integrates with current diagnostic workflows and what happens post-test for each risk category.
  5. The manuscript is generally well-written, but several sentences (especially in the abstract and results section) could benefit from stylistic refinement and grammar correction to improve readability and flow.
  6. While conflicts of interest are disclosed, the involvement of multiple authors employed by the test’s manufacturer warrants an additional statement on how data integrity and analysis objectivity were ensured.
  7. A brief comparison with existing non-invasive biomarker tests (PHI, 4Kscore, PCA3, IsoPSA) should be included to help contextualize the novelty and performance of the PSE test.

The manuscript will be suitable for publication after the issues above are clarified.

Comments on the Quality of English Language

The manuscript is generally well-written, but several sentences (especially in the abstract and results section) could benefit from stylistic refinement and grammar correction to improve readability and flow.

Author Response

(The authors gave the same response as above.)

Round 2

Reviewer 1 Report

Comments and Suggestions for Authors

The authors were improved their manuscript very well